# Colorectal Tumour Mucosa Microbiome Is Enriched in Oral Pathogens and Defines Three Subtypes That Correlate with Markers of Tumour Progression

**DOI:** 10.3390/cancers13194799

**Published:** 2021-09-25

**Authors:** Barbora Zwinsová, Vyacheslav A. Petrov, Martina Hrivňáková, Stanislav Smatana, Lenka Micenková, Natálie Kazdová, Vlad Popovici, Roman Hrstka, Roman Šefr, Beatrix Bencsiková, Lenka Zdražilová-Dubská, Veronika Brychtová, Rudolf Nenutil, Petra Vídeňská, Eva Budinská

**Affiliations:** 1Research Centre for Applied Molecular Oncology (RECAMO), Masaryk Memorial Cancer Institute, 656 53 Brno, Czech Republic; zwinsova@recetox.muni.cz (B.Z.); martina.hrivnakova@mou.cz (M.H.); hrstka@mou.cz (R.H.); sefr@mou.cz (R.Š.); bencsikova@mou.cz (B.B.); vebrychtova@mou.cz (V.B.); nenutil@mou.cz (R.N.); petra.videnska@recetox.muni.cz (P.V.); 2Research Centre for Toxic Compounds in the Environment (RECETOX), Faculty of Science, Masaryk University, 625 00 Brno, Czech Republic; viacheslav.petrov@recetox.muni.cz (V.A.P.); ismatana@mail.muni.cz (S.S.); lenka.micenkova@recetox.muni.cz (L.M.); mnau@mail.muni.cz (N.K.); vlad.popovici@recetox.muni.cz (V.P.); 3Institute of Biostatistics and Analyses, Faculty of Medicine, Masaryk University, 625 00 Brno, Czech Republic; 4Research Centre of Information Technology, IT4Innovations Centre of Excellence, Brno University of Technology, 601 90 Brno, Czech Republic; 5Department of Pharmacology, Faculty of Medicine, Masaryk University, 625 00 Brno, Czech Republic; lzd@mail.muni.cz; 6Department of Laboratory Medicine-Clinical Microbiology and Immunology, University Hospital Brno, 625 00 Brno, Czech Republic

**Keywords:** colorectal cancer, 16S rRNA gene, tumour microbiome, microbial subtypes

## Abstract

**Simple Summary:**

Dysbiosis of the gut microbiome may contribute to the heterogeneity of colorectal cancer from phenotypic, prognostic and response to treatment perspectives. We analysed CRC microbiome by 16S rRNA gene sequencing of paired tumour mucosa, adjacent visually normal mucosa and stool swabs of 178 patients with stage 0–IV CRC. We observed that tumour mucosa is dominated by pathogenic bacteria of oral origin and proposed a CRC tumour microbiome subtyping system. The subtypes and tumour mucosa genera were associated with prognostic clinical covariates (tumour grade, localisation, TNM, *BRAF* mutation and MSI). In contrast, changes in the stool microbiome were associated with lymph node involvement and the presence of synchronous metastases. We discovered new associations between microorganisms and CRC and clinical parameters. Our study represents a step forward in understanding the role of the microbiome and its interactions with factors involved in tumour progression, and it opens novel avenues for exploring new treatments and biomarkers.

**Abstract:**

Long-term dysbiosis of the gut microbiome has a significant impact on colorectal cancer (CRC) progression and explains part of the observed heterogeneity of the disease. Even though the shifts in gut microbiome in the normal-adenoma-carcinoma sequence were described, the landscape of the microbiome within CRC and its associations with clinical variables remain under-explored. We performed 16S rRNA gene sequencing of paired tumour tissue, adjacent visually normal mucosa and stool swabs of 178 patients with stage 0–IV CRC to describe the tumour microbiome and its association with clinical variables. We identified new genera associated either with CRC tumour mucosa or CRC in general. The tumour mucosa was dominated by genera belonging to oral pathogens. Based on the tumour microbiome, we stratified CRC patients into three subtypes, significantly associated with prognostic factors such as tumour grade, sidedness and TNM staging, *BRAF* mutation and MSI status. We found that the CRC microbiome is strongly correlated with the grade, location and stage, but these associations are dependent on the microbial environment. Our study opens new research avenues in the microbiome CRC biomarker detection of disease progression while identifying its limitations, suggesting the need for combining several sampling sites (e.g., stool and tumour swabs).

## 1. Introduction

Colorectal cancer (CRC) is the third most frequent cancer worldwide and the second leading cause of cancer mortality in Europe [1]. It is a heterogeneous disease, both from a phenotypic and a prognostic and response to treatment perspective. The current standard treatments are limited and remain ineffective for many CRC patients due to inadequate patient selection, resulting in unneeded toxicity and high cost resulting from over-treating of patients that do not benefit [2,3]. Recent research shows that gut microbiota may significantly influence colorectal tumour initiation and progression [4,5,6,7,8,9,10,11,12,13,14,15,16,17,18,19,20,21,22].

Several studies showed that bacteria adherent to colorectal adenomas or carcinomas were different from bacteria adherent to healthy gut mucosa [8,11,12] due to the altered tumour environment with decreased pH and modified metabolic conditions resulting from hypoxia and onset of necrosis [23]. Gut microbiota can promote colon cancer development or change the tumour invasion potential through (i) immunomodulation [10,24,25,26] or (ii) metabolic activity—via the production of specific toxins inducing DNA damage responses. Overall, the evidence of microbiome importance in colon cancer development is so overwhelming that a bacterial driver-passenger model for colorectal cancer development and progression has been suggested [27] as an alternative to the universally accepted driver-passenger mutational adenoma-carcinoma model. Additionally, gut microbiota seems to play a crucial role also in response to anti-cancer therapy [28].

Previous studies associating gut dysbiosis with CRC were focused on comparing the gut microbiome in the normal-adenoma-carcinoma sequence [4,5,6,7,8,9,10,11,12,13,14,15,16,17,18,19,20,21,22,29,30,31,32]. It is the landscape of the microbiome within the ongoing disease and its associations with clinical variables that remain under-explored. The published studies vary in techniques employed, specimen origin and sample size, thus hampering any integrative analysis. Most studies compared diseased and healthy subjects, and the few that tried to characterise microbial composition within the CRC patients suffered from a small sample size. The specimens used in most studies were stool [4,6,7,15,17,18,20,21,22] or mucosa samples from colonoscopy biopsies [11,13,15] or post-resection [6,12,16,19]. Stool microbiota sampling has the advantage of being non-invasive, allowing its use for screening and follow-up studies. Some efforts combined information about the tumour-associated microbiome with existing prognostic scores in an attempt to improve the prediction accuracy [18] or to develop a new screening/prognostic model [33]. The results of two different meta-analyses showed that the accuracy of predicting diseased state was about 0.8, such as occult blood test results, the main non-invasive clinical test for this type of cancer [34,35]. However, the microbial composition in stool only partially reflects the situation in tumour mucosa, a trend consistent across different nationalities of the patients, sampling techniques or sequencing methodology [36].

The microbiota adherent to the mucosal tissue differs from the faecal microbiota in its needs for oxygen and nutrient types [37,38]. Therefore, the information derived from stool may be insufficient for capturing tumour-microbe interactions consistent with the disease prognosis. The relevance of the tumour mucosa microbiome assessment for screening purposes is dependent not only on the co-presence of the bacteria in both tumour mucosa and stool but also on its association with relevant clinical parameters in both sample types. Additionally, studying the (dis)similarity of bacterial composition between tumour and visually normal mucosa from the same individual may provide hints regarding the changes in microenvironment which have occurred favouring the growth of certain species and shed some light on the underlying tumour-immune system-microbe interactions and metabolic pathways.

Recently, two studies provided a comparison of bacterial composition in both tumour tissue and visually normal tissue and the bacterial composition of stool samples from the same patients [34,35]. Liu et al. [34] showed that the bacterial communities in both tumour tissue and visually normal tissue were similar. Still, the study was vastly underpowered (*n* = 8 individuals) and did not explore the clinical relevance of this similarity. Other studies associated microbiome on tumour or in stool with clinical variables [35,39,40] but had a similar disadvantage in terms of statistical power (*n* = 25, *n* = 30, *n* = 53, individuals, respectively).

The studies mentioned were species-centric because they compared the abundance of individual microbial species between the groups of interest. However, a broader view is needed to account for lesser-known species coupled with a larger sample size allowing for capturing enough inter-tumour heterogeneity, thus better understanding the possible effects of bacteria on tumour growth, aggressiveness or response to therapy. Our study takes a microbial community-centric approach to provide a comprehensive description of the CRC tumour microbiome based on 16S rRNA sequencing. We analyse three sample types (tumour mucosa, visually normal mucosa, stool) from *n* = 178 individuals with stage 0–IV colorectal cancer.

Our study has a dual nature, both exploratory and confirmatory. We explore and interpret the landscape of the tumour mucosa-associated microbiome with respect to clinical variables and microbial composition of paired adjacent visually normal mucosa and paired stool samples. Benefitting from a larger sample size, we advance the state-of-the-art knowledge by reporting previously unseen associations. Most importantly, we capture the tumour microbial heterogeneity and derive CRC tumour microbiome subtypes.

## 2. Materials and Methods

### 2.1. Patients and Specimens

All specimens were collected at Masaryk Memorial Cancer Institute (Brno, Czech Republic) from 2015 to 2019. Patient inclusion criteria were (i) scheduled for resection based on preliminary screening (such as a colonoscopy), (ii) no neoadjuvant treatment, (iii) no previous CRC diagnosis (iv) with confirmed stage 0–IV CRC without multiplicities (single tumour). The stool samples were collected from untreated patients before the scheduled surgery. Patients performed the collection at home, the morning of their hospitalisation for the surgery and brought the samples to the hospital, where they were immediately frozen at −80 °C until further processing. Swabs from the tumour and visually normal mucosa were collected within 30 minutes of the tumour resection at the pathology department. Whenever possible, the swab from visually normal tissue was taken at least 20 cm proximally to the tumour. The swabs were then stored immediately in a freezer at −20 °C and, without unnecessary delay, transferred to −80 °C until further processing. All samples, including stool, were collected using DNA free cotton swabs (Deltalab, Barcelona, Spain).

Overall, we analysed *n* = 483 samples from *n* = 178 CRC patients. There were 127 triplets (all three sample types from the same patient) and 51 mucosa duplets (swabs from tumour and visually normal mucosa from the same patient).

The study was approved by the ethical committee of Masaryk Memorial Cancer Institute. All patients gave written informed consent following the Declaration of Helsinki prior to participating in the study.

### 2.2. DNA Extraction, PCR Amplification and Sequencing of 16S rRNA Gene

According to the manufacturer’s instructions, the DNA extraction was performed using DNeasy^®^ PowerSoil^®^ Isolation kit (QIAGEN, Düsseldorf, Germany). Extracted DNA was used as a template in amplicon PCR to target the V4 hypervariable region of the bacterial 16S rRNA gene. The 16S metagenomics library was prepared according to the 16S Metagenomic Sequencing Library Preparation protocol (Illumina, San Diego, CA, USA), with some deviations described below. Each PCR was performed with HotStarTaq Master Mix Kit (QIAGEN, Hilden, Germany) in triplicate, with the primer pair consisting of Illumina overhang nucleotide sequences, an inner tag, and gene-specific sequences [41,42]. The Illumina overhang served to ligate the Illumina index and adapter. Each inner tag, i.e., a unique sequence of 7–9 bp, was designed to differentiate samples into groups. Primer sequences and PCR cycling conditions are summarised in Appendix A. After PCR amplification, triplicates were pooled, and the amplified PCR products were determined by gel electrophoresis. PCR clean-up was performed with Agencourt AMPure XP beads (Beckman Coulter Genomics, Danvers, MA, USA). Samples with different inner tags were equimolarly pooled based on fluorometrically measured concentration using Qubit^®^ dsDNA HS Assay Kit (Invitrogen™, Carlsbad, CA, USA) and microplate reader (Synergy Mx, BioTek, Winooski, VT, USA). Pools were used as a template for a second PCR with Nextera XT indexes (Illumina, USA). Differently indexed samples were quantified using the qPCR kit KAPA Library Quantification Complete Kit (Roche, Indianapolis-Marion County, IN, USA) and LightCycler 480 Instrument (Roche, USA) and equimolarly pooled according to the measured concentration. The prepared libraries were checked with a 2100 Bioanalyzer Instrument using the High Sensitivity D5000 Screen tape (Agilent Technologies, Santa Clara, CA, USA), and concentration was measured with qPCR shortly prior to sequencing. The final library was diluted to a concentration of 8 pM, and 20% of PhiX DNA (Illumina, USA) was added. According to the manufacturer’s instructions, sequencing was performed with the Miseq reagent kit V2 (500 cycles) using a MiSeq instrument (Illumina, USA).

### 2.3. Data Analysis

#### 2.3.1. Preprocessing and Quality Control

Forward and reverse pair-end reads were demultiplexed, and barcodes and primers were trimmed. Denoising algorithm with DADA2 [43] was applied separately on forward and reverse reads that passed the quality and length filter and did not contain N’s. Reads were merged using the fastq-join method [44]. In the next step, chimaeras were detected with the function removeBimeraDenovo in DADA2. Chimaera sequences were subsequently excluded from the analysis, and Amplicon Sequence Variant (ASV) table was created.

After quality filtering and chimaeras removing, the number of reads ranged from 2968 to 239,116, with a median of 44,371 and a mean of 53,074 reads per sample. The number of reads did not differ between the sample types (paired Wilcoxon test, Appendix A).

#### 2.3.2. Taxonomy Assignment and Metabolic Potential Prediction

Taxonomy was assigned to each ASV based on SILVA 123 reference database [45] using the algorithm UCLUST [46] in QIIME [47]. BLAST algorithm [48] was used to identify the species, and all taxa with the maximum identity and minimum *e*-value were selected for each ASV. The observed species metric and the Chao1 and Shannon index were used to estimate alpha diversity for each sample in QIIME. Beta diversity was computed in QIIME using both weighted and unweighted UniFrac metrics [49].

We filtered out the ASVs unassigned at the phylum level and all the ASVs belonging to the phylum Cyanobacteria. Only the taxa present in at least three samples of the same sample type and at the same time represented by at least nine reads were kept for further analysis to account for possible contaminations. The threshold of 9 reads represents 0.3% taxa abundance in the sample with the least number of reads (2968).

This filtering step discarded 46–55% of taxa at each taxa level (Appendix A).

Picrust 2 [50] was used to predict hypothetical abundances of *KEGG* orthologs in each sample and to summarise them into higher functional processes.

#### 2.3.3. Statistical Analysis and Data Mining

All comparisons between the three sample types were performed on triplet samples from 127 patients, totalling *n* = 381 samples for the analysis. For the analysis of tumour-visually normal mucosa pairs, we used paired tumour and visually normal mucosa swabs from 178 patients (totalling 356 samples). We used all the available samples for analyses performed within each sample type (178 for tumour mucosa swabs, *n* = 178 for visually normal tissue mucosa swabs and *n* = 127 for stool).

Data were analysed using appropriate corrections and approaches for compositional data [51,52,53,54]. Zero multiplicative replacement [53] was applied prior to the centred log-ratio (clr) transformation.

Non-metric Multidimensional Scaling (R vegan package [55]) over Aitchinson distance matrices (R coda.base package [56]) was used to analyse tumour microbial heterogeneity and β-diversity. To estimate the contribution of clinical traits in the microbiome, β-diversity permutational multivariate analysis of variance for distance matrices (R adonis function of vegan 2.5.4 package [55]) with 999 permutations were used. To assess the differences between the sample types in alpha diversity, we used a paired non-parametric two-way Mann-Whitney *U* test. We applied a non-parametric approach to identify differences in microbial composition between sample types and the associations between relative microbial abundance and clinical variables. For non-parametric analysis, the Friedman test with paired Wilcoxon test and rank regression was used (R package Rfit [57]). A drop in dispersion test was used to produce overall *p*-values for rank regression models. The Cochran *Q* test was used to analyse differences in the presence of genera across sample types (analysis of triplets). Benjamini-Hochberg correction for multiple hypothesis testing was applied [58]. Results were considered significant at FDR <0.1. The adjusted *p*-values are referred to as *q*-values. Visualisation was performed with gplots 3.0.1.1, ggplot2, ComplexHeatmap 1.17.and circlize 0.4.8 packages [59,60,61,62].

For each clinical variable (or a combination thereof), we only tested genera present in at least 10 samples in one clinical group (or a combination thereof). We do emphasise that we approached this statistical testing from the point of view of a pilot discovery study.

Due to the known association between tumour grade and location [63] (also confirmed in our data, *p* < 0.001, Fisher’s exact test), we investigated the associations of the microbiome with grade and tumour location in a model with the interaction between covariates compared to a model without interaction. To ensure a more balanced design, we considered three locations: right and transverse, left, rectosigmoid and rectum, respectively.

The threshold of false discovery rate was set to 0.1, as is customary in similar studies, with the aim to identify potential candidates for further research. While we consider only associations with FDR <0.1 to be statistically significant, we also report the unadjusted results *p* < 0.05 for hypothesis confirmation by other studies.

### 2.4. Data Access

The data were uploaded to the European Nucleotide Archive under accession number PRJEB35990.

### 2.5. Validation

We performed partial validation of our results on three publicly available datasets. The association of the tumour microbiome with tumour localisation was validated in the dataset of Dejea et al. [31], *n* = 23. No grade information was available, and hence in the validation we did not use the grade*localisation interaction term. Publicly available fastq files were analysed with QIIME pipeline with the appropriate approach for 454 Roche sequencing. Taxonomy was assigned using SILVA 123 database to have comparable results with our dataset.

The association of stool microbiome with AJCC staging and TNM staging was validated in two datasets (Zeller et al. [32] and Feng et al. [30]). The processed datasets with taxonomic information were used as available in R package curatedMetagenomicData [64] and were normalised using the clr transformation before the analysis.

All associations were tested using rank regression (R package Rfit [57]). The dataset of Feng et al. only contained one M1 sample; hence we only analysed associations with AJCC staging, T stage and N stage.

## 3. Results

In our effort to describe tumour microbial landscape, we explored the differences in microbiome abundance, diversity, the presence/absence of the species and the proportion of samples with the respective genera in different sample types across patient groups defined by clinical variables (Table 1).

### 3.1. Microbial Categorisation According to Sample Type

There was no significant difference between the read counts across different sample types (paired analysis of sample triplets, see Methods).

The analysis of the 127 triplet samples revealed that the microbial diversity was significantly decreased in mucosal samples (both tumour mucosa and visually normal mucosa swabs) compared to stool, as measured by the number of observed species, Chao 1 and Shannon index (Appendix A). No differences were found between the tumour mucosa swabs and visually normal mucosa swabs.

Overall, in all the 483 samples, we identified 5449 ASVs: of these, 4800 ASVs in the 127 triplet samples. The QIIME assigned species only to 48 ASVs. Hence, we also performed a manual BLAST search to the SILVA database (Appendix A).

For further analysis, however, we operated on higher taxonomic levels. After the taxa filtering step (Appendix A), 13 phyla, 25 classes, 43 orders, 75 families and 264 genera were identified in the 127 triplets, most of which in all three sample types (Appendix A). Inclusion of the additional 51 duplets (tumour mucosa and visually normal mucosa swabs) resulted only in slight differences at the genus level—the identified taxa remained the same. What changed was their unique presence in some sample types (Appendix A).

While most of the genera were found in all three sample types, their incidence and abundance across sample types varied greatly between mucosal samples and stool, both in overall and pairwise comparisons (Appendix A). In this case, 14 genera (*Stomatobaculum, Pseudoramibacter, Pelomonas, Pasteurella, Mycoplasma, Kingella, Johnsonella, Helicobacter, Deinococcus, Centipeda, Bergeyella, Actinobacillus, Abiotrophia* and an unassigned genus from order *Comamonadaceae*) were detected only in mucosal (tumour and visually normal) samples (Appendix A).

We further analysed the pairwise incidence of the 264 genera across sample types. We found that 104 genera varied significantly across sample types (analysis of 127 triplets, Appendix A).

To categorise the microbial genera based on their preferred environment: we compared their abundance across sample types. Of the 264 genera, 121 differed significantly in abundance across the sample types (Appendix A). Based on these results, we defined five microbial categories (Figure 1). The first is based solely on the results of tumour vs stool comparison: tumour genera (57 genera, more abundant in tumours than stool). Additionally, within the category of tumour genera, we defined mucosa genera (52 genera, also enriched in visually normal mucosa compared to stool) and tumour-specific genera (16 genera of tumour category, additionally enriched in tumours compared to visually normal mucosa). In this case, 49 genera were significantly more abundant in stool than tumours and visually normal mucosa from the group of stool genera. The fifth category was defined as the no-difference genera (143 genera, no difference across any of the sample types) (Appendix A).

### 3.2. The Landscape of CRC Tumour Microbiome

For the description of tumour mucosa microbial heterogeneity without stool contaminants, we only considered species that were statistically significantly enriched in tumour mucosa compared to stool. We hence investigated the group of 57 tumour genera with a special focus on the subgroup of 16 tumour-specific genera (*Gemella, Granulicatella, Parvimonas, Hungatella, Peptoclostridium, Flavonifractor, Selenomonas 3, Fusobacterium, Leptotrichia, Eikenella, Campylobacter, Slackia, Streptococcus, Howardella, Solobacterium, Defluviitaleaceae UCG-011,* Figure 2A).

We performed the analysis of co-occurrence and observed significantly increased co-occurences between 20 tumour genera (of which 13 tumour-specific) (Appendix A). We also observed 14 significantly decreased co-occurrences between genera (Appendix A).

Tumour genera incidence ranged from 1.1% to 99.4% (median 26.4%) of tumours with the median abundance of the individual genera in the samples with the genus detected ranging from 0.01% to 29.8% (median 0.15%) (Figure 2A). Overall, tumour genera constituted 1.1% to 97% (median 59.6%) while the tumour-specific genera constituted between 0.0–62.3% (median 3.1%) of the microbiome found on tumour mucosa (Figure 2C).

We performed a detailed literature search (Appendix A) which revealed that tumour genera consisted predominantly of oral bacteria, many known as oral pathogens.

Some of the tumour genera of (possible) oral origin identified in our study, while previously associated with CRC, were never reported on tumour mucosa, namely *Solobacterium* (increased in CRC faecal samples [35])*, Slackia* and *Pseudomonas* (decreased [12,19] in CRC faecal samples), and *Treponema* (the presence of which in the oral cavity was associated with increased risk of CRC [33]).

We newly identified many genera of both oral and gut origin, not previously associated with CRC, with increased abundance in the tumour lesions: *Selenomonas 3, Selenomonas 4, Aggregatibacter, Actinobacillus, Bergeyella, Phocaeiola, Defluviitaleaceae UCG-011, Abiotrophia, Johnsonella, Stomatobaculum, Kingella, Shewanella, Tatumella, Senegalimassilia, Aeromonas, Prevotellaceae UCG-003, Incertae Sedis* genus from family *Erysipelotrichaceae,* an uncultured species from *Veillonelaceae* family, and an uncultured species from *boneC3G7* at the family level (BLAST hit *Fusobacterium necrophorum*) (oral origin) and *Tyzzerella 4, Massilia* and an unassigned genus from *Peptostreptococcaceae* family (gut origin).

Amongst tumour genera of gut origin, *Lachnoclostridium*, *Flavonifractor* [65,66], *Sutterella* and *Hungatella* (ex-*Clostridium hathewayi*) [67] were previously only reported increased in the stool of patients with CRC.

### 3.3. Microbiome and Clinical Variables

Prior to the subtype derivation, we assessed the association of bacterial genera from all the sampled environments with the clinical parameters and interpreted the results based on our microbial categorisation. We performed partial validation on three publicly available datasets.

β-diversity analysis by NMDS performed on each sample type separately showed that tumour location was the factor with the highest influence on total microbiome composition for all sample types, while tumour histological grade affected only tumour samples (Appendix A).

The results of regression analysis for each clinical variable are summarised in Table 2 and Appendix A; the detailed results of partial validation are presented in Appendix A.

Increased abundance of *Fusobacterium, Campylobacter* and *Leptotrichia* in tumour mucosa appeared to be independent predictors of tumour’s higher grade (*p* < 0.01, FDR < 0.1). *Leptotrichia* was significantly increased on visually normal mucosa adjacent to grade 3 left-sided tumours (*p* < 0.05, FDR < 0.1).

The mucosa of grade 3 right-sided tumours was enriched in *Prevotella, Selenomonas* and *Selenomonas 3* (*p* < 0.01, FDR < 0.1). *Prevotella* was also increased in the stool of patients with grade 3 rectosigmoid/rectum tumours (*p* < 0.01, FDR < 0.1). The mucosa of grade 3 tumours of the rectosigmoid/rectum and visually normal mucosa adjacent to left, rectosigmoid and rectal tumours, regardless of the grade, were enriched in *Lachnospira* (*p* < 0.05, FDR < 0.1).

The mucosa of left-sided (for some including rectosigmoid/rectum) low-grade tumours was enriched in *Ruminiclostridium 6, Coprococcus 2*, [*Eubacterium] ventriosum* group, *Clostridiales Vadin BB60* group, *Ruminococcaceae UCG-010* and an uncultured species and an *Incertae Sedis* genus from the *Lachnospiraceae* family (*p* < 0.01, FDR < 0.1)*. Ruminoclostridium 6* remained enriched also in the stool of patients with grade 2 left-sided, rectosigmoid and rectal tumours (*p* < 0.01, FDR < 0.1). *Methanobrevibacter*, *Victivallis* were significantly enriched in the mucosa of low-grade tumours of rectosigmoid and rectum (both *p* < 0.01, FDR < 0.1).

*Christensenellaceae R-7 group*, *Bifidobacterium* and *Ruminococcaceae UCG-013* were increased in mucosa of the left-sided, rectosigmoid and rectal tumours (*p* < 0.01, FDR < 0.1). Similar associations were found for visually normal mucosa for *C**hristensenellaceae R-7 group*, *Coprococcus 1*, *Lachnospira* and *Bifidobacterium* (*p* < 0.01, FDR < 0.1). The increased abundance of the *Christensenellaceae R-7* group in tumour mucosa of left-sided tumours was also validated in an independent dataset (*p* = 0.0047) (Appendix A). When comparing early (0–II) and advanced (III–IV) stages, we identified an increased abundance of *Akkermansia* in the stool of advanced stage tumours (*p* < 0.01, FDR < 0.1) (Appendix A).

Patients with advanced T stages (pT 3–4) were characterised by a significant increase in abundance of *Gemella, Campylobacter, Peptoclostridium and Parvimonas* (*p* < 0.01, FDR < 0.1) on tumour mucosa, and increased *Peptoclostridium, Escherichia-Shigella* (*p* < 0.01, FDR < 0.1) in the adjacent visually normal mucosa. Early T stage tumours (pTis-2) were associated with an increase in *Coprobacter*, on tumour mucosa, increased *Intestinimonas*, *Ruminococcaceae UCG-009*, *Holdemanella* and *Coprobacter* on the adjacent visually normal tissue (*p* < 0.05, FDR < 0.1) and *Prevotella 6* (*p* < 0.01, FDR < 0.1) and *Ruminococcaceae UCG-011* (*p* < 0.05, FDR > 0.1) in the stool (Appendix A). We validated the decrease of *Ruminococcaceae* in the stool of patients with pT 3–4 stage (*p* = 0.00113, Feng et al.) (Appendix A).

The presence of metastases (local or distant) at the time of diagnosis was predominantly associated with changes in the stool microbiome. Except for the increased abundance of *Akkermansia* in stool of patients with N1–2 stage tumours (*p* < 0.01, FDR < 0.1) and of the uncultured genus from the *Erysipelotrichaceae* family in the stool of patients with synchronous distant metastases (*p* < 0.01, FDR < 0.1), none of these associations were significant after FDR correction (Appendix A). Nevertheless, we validated the decrease of *Dorea* in the stool of patients with N1–2 stage tumours (*p* = 0.00011) in an independent dataset (Feng et al.) (Appendix A).

### 3.4. Tumour CRC Microbial Subtypes

We continued our characterisation of tumour microbial heterogeneity by performing hierarchical clustering of patients based on the relative abundance of the 57 tumour genera in the tumour mucosa samples (See Methods). Once the subtypes were identified, we performed between-subtype differential abundance analyses of microbiome profiles in all three sampling environments.

Based on the tumour genera profiles, we observed three major subtypes of tumours (TMS1–TMS3), that could further be divided into two groups each (Figure 3 and Figure 4). The bacteria were clustered into six groups B1–B6 (Figure 3, Appendix A).

The B1 group and B2 group are represented by typical gut microbiome members. The B1 group consists of the five most common and most abundant genera *Fusobacterium*, *Lachnoclostridium, Bacteroides, Escherichia-Shigella* and one uncultured genus from the family *Lachnospiraceae.* All tumours contain at least three of these bacteria, most tumours (78.7%) all five. These bacteria have high co-occurrence across the sampled environments (Figure 2A, fourth panel), except for *Fusobacterium,* predominantly found in mucosa samples. The B4 group contains exclusively oral microbiome genera, and we named it the *Selenomonas* group due to its enrichment in the *Selenomonas* genera. B3 and B5 groups include mostly oral microbiome genera. These genera have significantly different incidence across the sampled environments, with 45.7–94.1% of patients missing these genera in the stool if present on tumour mucosa. Group B6 consists of 27 less common species with incidence ranging from 0% to 37% (median 11.1%).

Tumour microbial subtype 1 (TMS1) represents 26% (46) of tumours and is defined by the presence of B1–B4 microbial groups, and overall contains most of the high-grade associated genera *(Fusobacterium, Campylobacter, Leptotrichia, Peptoclostridium* and *Selenomonas,* see Table 2*).* This subtype is enriched in right-sided (60.9%), grade 3 (58.7%), pT3 or pT4 stage (95.6%) tumours and is depleted of stage 0 and stage I tumours (0% and 4.3%, respectively) (Table 1, Figure 4). In addition, TMS1 contains significantly more tumours with MSI-H (34.8%) and BRAF mutation (15.2%) compared to other tumour microbial subtypes. TMS1 differs from TMS2 and TMS3 by the presence of the *Selenomonas* group *(B4), Solobacterium* and *Howardella* species, and *Clostridium sensu stricto 1*. In contrast to other subtypes, this subtype shows a significantly decreased abundance of typical faecal commensals such as *Bifidobacterium, Ruminococcus 2, Anaerostipes* and *Coprococcus 1* on tumour mucosa (Appendix A). In stool samples, we observed a higher abundance of *Prevotella* and *Clostridium sensu stricto 1* (Appendix A).

Tumour microbial subtype 2 (TMS2) comprises 31% (55) of tumours and is defined mainly by the absence of B4 bacteria (the *Selenomonas* group). This subtype can be further divided into two groups by the increased incidence of *Leptotrichia, Granulicatella, Aggregatibacter* and *Neisseria* (TMS2a) or *Tyzzerella 4, Hungatella (*ex*-Clostridium hathewayi), Solobacterium, Pseudomonas* and *Porphyromonas* (TMS2b). TMS2 tumours are predominantly from the left side, rectosigmoid or rectum (70.9%) (Table 1, Figure 4). The mucosa of TMS2 tumours shows a significantly higher abundance of *Haemophilus*, *Sutterella*, *Veillonella* and *Streptococcus* and a lower abundance of *Alloprevotella* (Appendix A). *Alloprevotella* was also significantly decreased in stool samples (Appendix A).

Finally, the largest subtype, TMS3, represents 43% (77) of tumours and is mostly missing the B3–5 bacterial groups and most of the high-grade related species. TMS3 is characteristic by an increased proportion of grade 1 tumours (15.6%). In the TMS3 microbial subtype, right-sided and left-sided tumours are equally represented (Table 1, Figure 4). The subtype can be further divided by increased incidence of *Incertae sedis* from the *Erysipelotrichaceae* family and *Tyzzerella 4* (TMS3a) or *Clostridium sensu stricto 1, Ruminococcaceae UCG-013* and *Incertae sedis* from *Lachnospiraceae* family (TMS3b). Interestingly, subtype TMS3 contains all the tumours that lack *Fusobacterium* species (most of them in TMS3a) both in their mucosa and in the patients’ stool.

Most importantly, the subtypes differed significantly in the proportion of the oral genera with TMS1 median of 15.8%, TMS2 median of 12.3% and TMS3 median of 5.3% (*p* < 0.001). We then explored the estimated metabolic potential of the microbial communities specific to the tumour subtypes (Appendix A). TMS1 is characterised by functional shifts in bacterial composition, including the increase in gene content specific for nucleotide metabolism, metabolism of terpenoids and polyketides and energy metabolism (*p* < 0.05, FDR < 0.1), reduction in lipid metabolism and xenobiotics biodegradation and metabolism (*p* < 0.05, FDR < 0.1). At the lower functional level, TMS1 subtype was characterised by increased peptidoglycan biosynthesis, novobiocin biosynthesis and ansamycin biosynthesis (*p* < 0.05, FDR < 0.1). The TMS2 subtype exhibited the highest score of xenobiotics biodegradation and metabolism. TMS3 subtype showed enhanced biosynthesis of other secondary metabolites, carbohydrate metabolism and amino acid metabolism and reduced metabolism of other amino acids (*p* < 0.05, FDR < 0.1).

## 4. Discussion

Carcinogenesis of colorectal cancer is a complex process with a unique set of somatic molecular changes. Considerable efforts have been dedicated to understanding the heterogeneity of CRC and deriving clinically applicable molecular markers of the disease progression and patients’ response to therapy. Approaching the problem from the molecular perspective in supervised analyses led to identifying several molecular markers and signatures with limited clinical use [73]. The unsupervised approach led to the definition of four consensus molecular subtypes [74], which, surprisingly, bear some prognostic value. The CRC heterogeneity puzzle, however, is far from being solved. One of the reasons is that the tumour microenvironment, especially the microbiome, seems to play a much more critical role than imagined. Many studies correlated the dysbiosis of the gut microbiome with the development of colorectal cancer in the healthy mucous-adenoma-carcinoma sequence or focused on elucidating the concrete role of selected bacterial species in the gut (colorectal) pathogenesis progression [4,5,6,7,8,9,10,11,12,13,14,15,16,17,18,19,20,21,22].

In contrast, our study aimed to use an unsupervised approach to characterise the heterogeneity of the CRC gut microbiome in the ongoing disease to discover unforeseen patterns. The comparison of microbial communities of stool, tumour mucosa and adjacent visually normal mucosa provided us with insights into the preferred environment of the observed species. The resulting microbial categorisation served to focus downstream analyses and to interpret our findings. We based the characterisation of the CRC tumour microbial landscape by performing subtyping of patients on bacteria with increased abundance in tumour mucosa compared to the stool to filter out potential stool contaminants. With a median of 59.6%, the tumour genera represented an essential fraction of the total microbiome found on tumour mucosa. Interestingly, the tumours of different microbial subtypes differed in the on-mucosa abundance of typical faecal genera (both pathogenic and commensal) that were not used for their definition. The analysis of microbial composition between sample types confirmed previously reported observations [8,11,34] that mucosa-associated bacteria dominate the tumour mucosal microbiome and that these species are associated also with visually normal mucosa. It is debatable to what extent the non-cancerous tissue (however distant from the tumour) from the surgically removed segment is already influenced by the bacteria initiating CRC development. Consistent with the bacterial driver-passenger model as proposed by Tjalsman et al. [27] our mucosa genera could be bacterial drivers, while tumour-specific genera could be bacterial passengers. We observed that tumours harbour a diverse community of opportunistic pathogens of oral origin (31 of 57 tumour genera) as previously reported [29,31,75].

Multiple factors make the CRC tumour niche a favourable environment for oral bacteria, in particular, for oral pathogens. Some of the bacteria can bind to specific proteins overexpressed on tumour cells [76,77,78]. Inflammation in the oral tissue niche selects for those species that are most adapted to the new environment, producing specific molecules such as microbial proteolytic enzymes [79] that break down the host’s extracellular matrix and soluble factors to get nutrients and invade the tissue. In the digestive tract, some oral bacteria can change their oxygen requirements from facultative anaerobic to strict anaerobic and their metabolism from asaccharolytic to proteolytic [80]. Oral pathogens gaining a more favourable niche on colon tissue may shift the balance on their behalf, producing proteins playing a key role in biofilm formation [81]. Some of the oral genera detected in our study were previously never associated with CRC tumour mucosa (e.g., *Selenomonas 3, Selenomonas 4, Aggregatibacter, Johnsonella, Abiotrophia, Defluviitaleaceae UCG-011*). Most importantly, we newly associate 22 genera of both oral and gut origin with CRC overall. Some of these genera contain species that are known human pathogens causing infections of mucosal or other tissues such as: periodontal disease (*Selenomonas, Phocaeiola Aggregatibacter)* [82,83,84] infections in humans through animal bites (*Bergeyella* and *Actinobacillus)* [85,86]; endocarditis (*B. cardium*) or respiratory infections (*A. hominis)* [87,88]. For other genera, their potential involvement in CRC is not so obvious. The tumour-specific genera of *Defluviitaleaceae* might influence CRC through the metabolism of butyrate [89]. The association of *Tyzzerella 4* from the *Lachnospiraceae* family with CRC may be due to its increased occurrence in patients with higher cardiovascular risk (CVR) factor scores [90], which are also associated with CRC [91]. *Massilia* was detected in patients with pancreatic cancer [92].

Correlating the tumour microbiome with clinical variables of tumour progression, such as grade or stage, bears the promise of offering viable hypotheses on the role of bacteria in the progression of the disease. Currently, the associations between clinical variables and gut microbial composition in an ongoing disease are understudied, and only a few efforts addressed the topic on limited cohorts.

The sample size of our study allowed for the study of the interaction of grade and tumour location, thus providing a finer estimation of the differences in the microbiome composition. We report 59 associations of 43 genera with tumour grade and/or location for all sample types studied. We confirmed previously reported high-grade tumour associations of *Fusobacterium, Campylobacter* and *Mogibacterium,* in CRC tumour mucosa [40,68]. We newly observed potentially beneficial effects of the increased abundance of 13 *stool genera* significantly associated with left location, namely decreasing tumour grade with increasing abundance, e.g., of *Bifidobacterium, Ruminococcaceae UCG-010* and *Victivallis* in tumour mucosa; and of *Porphyromonas and Lachnospiraceae UCG-005,* in the stool. *Bifidobacterium* was previously shown to have anti-cancerogenic effects [66,69,70,71,72].

We also found location-dependent grade-predictive genera. Remarkably, while in the right colon a higher grade was associated with an increase in pathogenic genera *(Prevotella, Selenomonas)* on tumour mucosa, in the left colon a higher grade was associated with a depletion in possibly beneficial (commensal) genera *(Methanobrevibacter, Coprococcus 2, Ruminiclostridium 6, Odoribacter, Dielma, Victivallis)* on tumour mucosa or in the stool. We can only speculate whether the prolonged exposure of tumour mucosa to predominantly stool bacteria that is mechanistically related to tumours in a distant part of the colon (left-sided or with onset in rectosigmoid and rectum) can have potentially harmful or beneficial effects or whether any associations are mostly due to the well-known molecular differences in the right vs left-sided tumours [36,93,94]. The increased abundance of pathogens on the high-grade right-sided tumours might be the result of increased permeability of proximal gut mucosa [95], but the relevance of the animal models was questioned [96].

We confirmed a previously published increase of *Akkermansia* and *Porphyromonas* in the stool of patients with local metastases [72], and newly associated increased *Akkermansia* in stool in patients with stage III–IV CRC. We noted that the occurrence of synchronous local and distant metastases was mainly associated with shifts in stool microbiome, while tumour specific variables such as grade or location were associated with changes in tumour microbiome. On one side, this observation raises the possibility of microbiome-based non-invasive metastasis diagnostics in colorectal cancer or monitoring the patients at risk. On the other hand, the alteration of the stool microbial community might only reflect changes in the overall health status in the presence of metastasis and cancer progression itself similarly to non-colonic malignancies [97,98,99].

Pairwise analysis of the incidence of all genera across sample types helped us to assess their screening potential. On-tumour microbes with significant clinical associations and no difference in incidence across sample types are perfect candidates for stool-based screening studies or stool-based prognostic and predictive classifiers. Most of the tumour-specific genera, if present on tumour mucosa, were not identified in stool of the same patients in more than 50% of cases. Given that these genera prefer the mucosal environment over the stool, such associations are not entirely surprising. Consequently, these genera are better candidates for colonoscopy biopsy sample screening.

We then compared how the previously suggested stool-based predictive microbial markers of CRC (compared to healthy and adenomas) [29] behave with respect to the progression of an ongoing disease (associations with grade or stage) as a result of tumour microenvironment changes. Some retained their predictive potential of progression of the disease as stool predictors of the presence of local metastases (increase in *Campylobacter, Porphyromonas, Streptococcus* and decrease in *Lachnospiraceae* and *Faecalibacterium*). Some showed no significant clinical associations in stool, but their increased abundance on tumour mucosa was predictive of high pT stage (*Parvimonas*) or grade (*Fusobacterium*).

The three tumour–mucosa-based microbial subtypes we derived on patterns of similarity of the abundance of the tumour genera represent the first attempt to systematically describe microbial heterogeneity of CRC tumour environment. We were intrigued to see that compared to subtyping efforts based on gene expression [74,100], also microbial profiling identified one subtype (TMS1) enriched in *BRAF* mutant, MSI-H, right-sided tumours. An interesting observation was that the tumour microbial subtypes differ not only in the type of the tumour genera they host but also in the count of potentially pathogenic microbiome correlated with high grade and stage and the proportion of oral pathogens within the tumour genera. Of the 10 high grade or high stage-related genera, TMS1 tumours had a median of 8 (80%), TMS2 of 6 (60%) and TMS3 of 4 (40%), differing thus in what we could call “microbial pathogens burden”. This subtyping could reflect differences in tumour biological properties linked with cancer progression: malignant tumours with active growth, cell and tissue atypia because of disruption of the mucus layer and dysregulation of local immunity provide more comfortable conditions to aggressive microbial consortia expansion and unconventional (oral) species homing. Moreover, with respect to the bacteria-supported model of carcinogenesis, proved in animal models [10,101], the pathogenic bacteria growth leads to additional dedifferentiation of tumour cells forming the pathogenetic loop. The differences in the proportion of oral pathogens and metabolic potential lead us to the hypothesis that the TMS1 subtype is enriched in tumours with microbial biofilms. This subtype is enriched in right-sided tumours and compared with the other two subtypes, is enriched in the presence of oral bacteria. Recently, biofilms have been associated with right-sided CRC [31]. Drewes et al. [102] identified several biofilm-associated shifts, including the functional alteration in peptidoglycan biosynthesis, novobiocin biosynthesis and ansamycin biosynthesis, which were significantly increased in the TMS1 subtype. Studies show that the commensal and the pathogenic periodontal bacteria *(Fusobacterium, Porphypomonas)* produce proteins such as gingipains [81] and RadD [103], which can play a key role in biofilm formation. Koliarakis et al. [75] proposed a new outlook on CRC pathogenesis driven by gut mucosa biofilm created by periodontopathic bacteria translocated into the colorectum. Tomkovich et al. [104] successfully demonstrated that polymicrobial biofilms are carcinogenic. Transcriptomic studies of periodontal tissues show that many organisms can fulfil gaps in metabolism, therefore the pathogenic community is more important as one unit than the virulence of one species [105].

It remains to be investigated whether the microbial subtypes could improve the prediction of patients’ survival and prognosis. We can speculate that the high microbial pathogen burden could worsen not only the tumour progression but also potentially the patient’s condition after the surgical resection and during and after the chemotherapy treatment. Given the fact that tumour-related genera reside also on visually normal mucosa, they could initiate CRC tissue dysplastic changes and malignisation. There is limited evidence of linkage between mucosal microbiota and metachronous adenomas growing demonstrated by Liu et al. [106]. On the other side, it is shown that the microbiome could interact and metabolise chemotherapeutic medicine, which leads to modulation of its activity and toxicity [107]. In the light of the above, modification of gut microbiome after colorectal cancer surgical removal might be considered as an additional step of treatment to prevent tumour recurrence and modulate chemotherapy effectiveness and toxicity.

The probable presence of biofilm in the TMS1 subtype might make this subtype of interest to potential prevention and treatment strategies. Importantly, although TMS1 is enriched in proximal tumours, it occurs in 9.1–18.8% of left, rectosigmoid and rectal tumours. Remarkably, biofilm communities from the colon biopsies of healthy individuals were as potent in inducing colon inflammation as the biofilm communities from CRC hosts [104]. The inhibition or removal of such biofilms from patients with CRC could represent a promising strategy for secondary CRC prevention and treatment but remains an uneasy task due to inefficiency of traditional antimicrobial strategies such as antibiotic treatment [108]. A recent study associated periodontitis with increased risk of high-grade proximal colorectal cancer [109]. Based on this, our results suggest an intriguing hypothesis: whether improving oral hygiene would impact the incidence of TMS1 tumours, or, more importantly, would lower the recurrence rate or development of secondary tumours in the TMS1 patients.

Another clinically relevant observation is the association of left-sided high-grade tumours with the depletion in protective species rather than increase in bacterial pathogens. This suggests that antibiotic treatment of patients with distal tumours may have a detrimental effect on their prognosis. Coadministration of probiotics in this case could be highly beneficial.

We believe that for certain patient populations, the inclusion of tumour mucosa sampling during colonoscopy for analysis of microbial composition could help to efficiently steer pre- and post-operative treatment decisions.

## 5. Conclusions

In our study, by analysis of 483 samples from *n* = 178 patients, we extended the current characterisation of the colorectal cancer microbiome in several directions. Thanks to the large sample size, we identified bacterial genera that were not previously associated with CRC tumour mucosa, clinical variables or with colorectal carcinoma at all. These genera should be studied in more detail to describe their mechanism of interaction with the disease.

By focusing on microbial community analysis, in contrast to classical microbiome-centred approaches, we were able to identify co-occurring species and three major tumour-microbial subtypes that correlate with clinical variables, such as grade, location and TNM staging. The subtypes also differ in what we describe as microbial pathogens burden—the number of pathogenic species correlated with increased grade and stage present on tumour mucosa, although the concept can be defined with respect to all three environments (tumour mucosa, visually normal mucosa and stool).

An important limitation of our study is the lack of proper validation of all the results since adequate data is unavailable, hence these results must be taken cautiously. Additionally, it is well known that the gut microbial composition changes with dietary patterns and lifestyle, which could be region-based [98]. More studies of similar sample size or larger from different geographical locations, are needed to derive robust and generalisable patterns. We make the full data available, including clinical variables, as a first step towards building a data corpus that could support such investigations. The nature of the samples (mucosa) prohibited us from using more advanced whole-metagenomic sequencing due to severe human DNA contamination issues [110,111,112]. The technology chosen was high throughput, fitting the purpose of microbial community-based analysis. We did perform the sequence matching for the identified ASVs against the SILVA database, however, being aware of the limitations, we provide these results solely as supplementary information without discussing them here in detail.

Having sampled the microbiome at three different complementary sites allowed the study of several environments leading to the definition of novel microbial categories with multiple implications. Our study shows that the associations with clinical variables found for the tumour mucosal or adjacent visually normal mucosa microbiome are rarely preserved in the microbial composition of stool and vice versa. While tumour histological grade, stage and location are reflected in the corresponding mucosal microbiome, the presence of lymph nodes or distant metastases influences mainly the stool microbiome. It seems that the mucosa and stool microbiome are complementary with respect to the modulation of their effects on disease progression. Tumour-mucosa biopsies from colonoscopy might need to be coupled with stool sampling for efficient screening or diagnostic purposes.

Understanding the role of tumour-subtype specific microbial communities could lead to tailored strategies of CRC patient gut microbiome management through lifestyle and diet recommendations including probiotic and antimicrobial interventions.

Our study is a step forward in understanding the role of the microbiome and its interactions with other factors involved in oncogenesis and tumour progression. Rather than providing definite answers it opens new avenues for exploring new treatments and biomarkers.

## Figures and Tables

**Figure 1 cancers-13-04799-f001:**
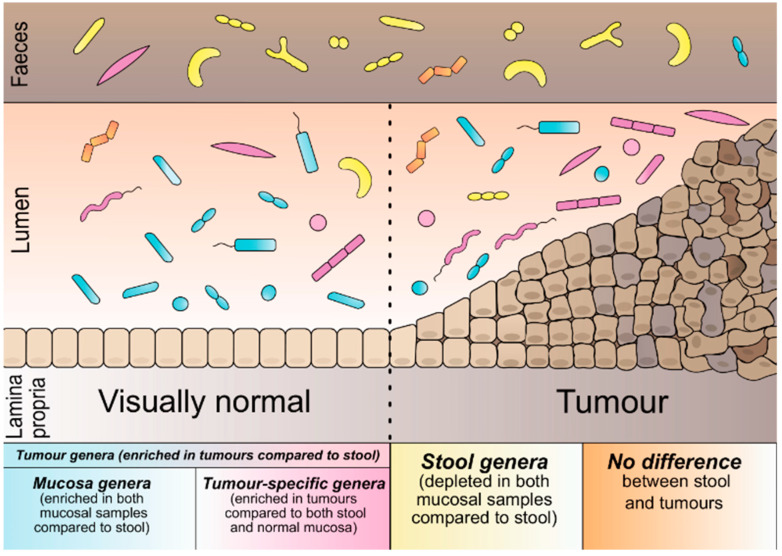
Schematic representation of bacterial categories according to their preferred environment.

**Figure 2 cancers-13-04799-f002:**
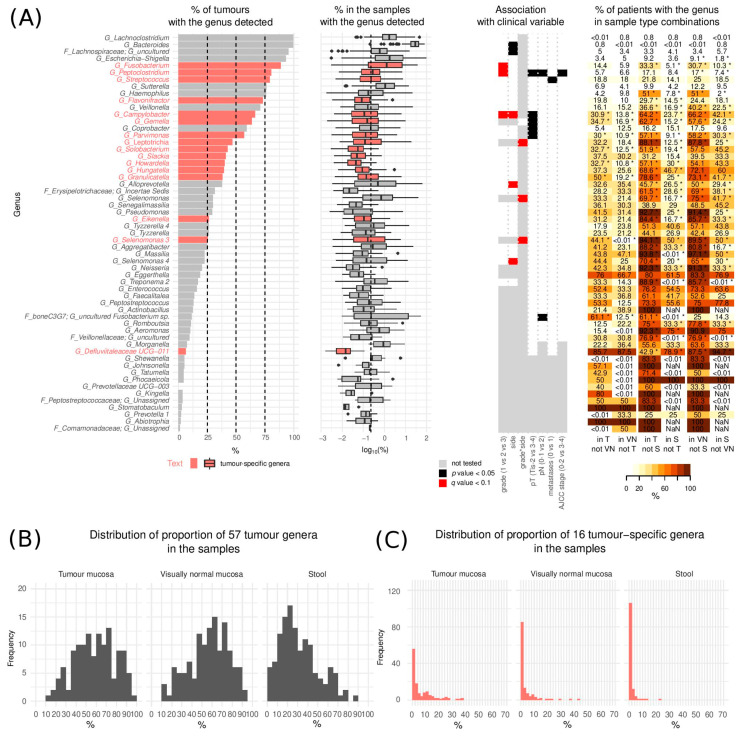
Tumour genera. (**A**) (left) Proportion of tumours with the genera detected and distribution of their respective relative abundances (in %) in the samples where they were detected (middle) and association of the bacteria with clinical variables (right). The vertical dashed line represents the median relative abundance of all 264 detected genera (median = 0.24%). The boxplot middle vertical line represents median, the box represents the interquartile range (IQR), the whiskers extend to +/−1.5 IQR. The black dots refer to outliers. (**B**) Overall proportion of the 57 tumour genera in the three sample types (*n* = 127) (**C**) Overall proportion of the 16 tumour-specific genera in the three sample types (*n* = 127). (Tis—Tumour in situ, pT—tumour pathologic stage, pN—regional lymph nodes pathologic, T—tumour swabs, VN—visually normal mucosa swabs, S—stool, NaN—not a number). * *p* < 0.05.

**Figure 3 cancers-13-04799-f003:**
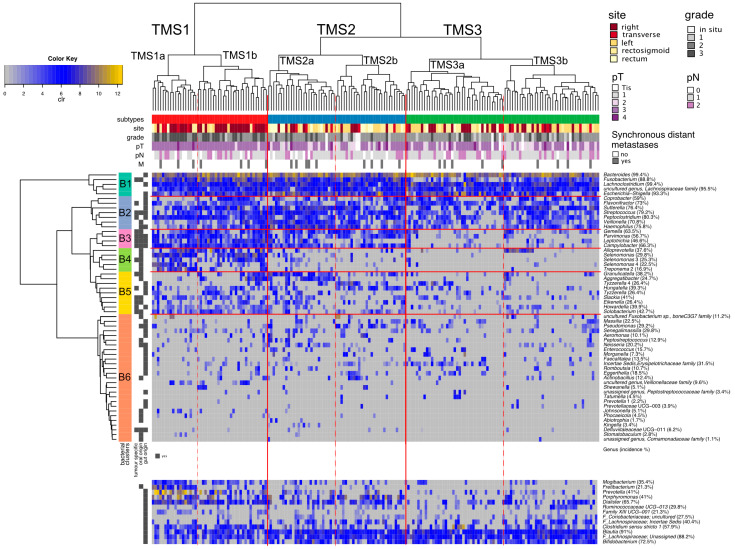
Tumour microbial subtypes. Hierarchical clustering of tumours (Aitchinson distance) and genera (Euclidean distance) based on the clr-transformed abundances of 57 tumour genera. Clinical variables of individual patients are shown. Proportions right to the genera name denote incidence of the genus in 178 tumour-mucosa samples. (TMS—tumour microbial subtypes, clr—centered log—ratio transformation, pT—tumour pathologic stage, pN—regional lymph nodes pathologic, M—synchronous distant metastasis, Tis—tumour in situ).

**Figure 4 cancers-13-04799-f004:**
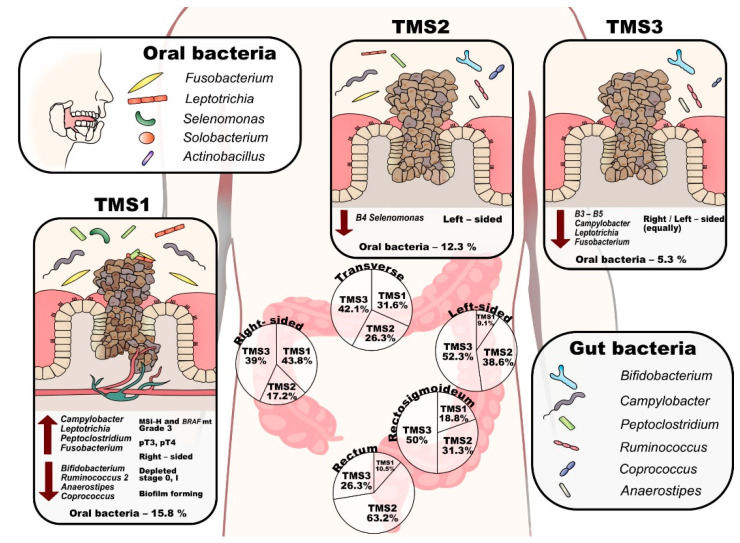
Scheme of the tumour microbial subtypes. (TMS—tumour microbial subtypes, pT—tumour pathologic stage, MSI-H— microsatellite instability-high).

**Table 1 cancers-13-04799-t001:** Table of clinical variables and their distribution in the complete set of 178 patients, the subset of 127 patients and in the CRC tumour microbial subtypes, respectively. (For categorical variable, Fisher exact test was performed and for continuous data, Kruskal-Wallis test was used.).

Clinical Variables	Data Subset Comparison			Tumour Microbiome Subtypes			
	All Tumours (*n* = 178)	Triplets (*n* = 127)	*p*-Value	TMS1 (*n* = 46)	TMS2 (*n* = 55)	TMS3 (*n* = 77)	*p*-Value
age at diagnosis	Mean (SD)	Mean (SD)	0.804	Mean (SD)	Mean (SD)	Mean (SD)	0.887
	66.92 (10.66)	66.61 (10.61)	-	66.89 (9.88)	67.47 (11.39)	66.55 (10.69)	-
gender	*n* (%)	*n* (%)	1	*n* (%)	*n* (%)	*n* (%)	0.729
male	99 (55.6)	70 (55.1)	-	25 (54.3)	33 (60.0)	41 (53.2)	-
female	79 (44.4)	57 (44.9)	-	21 (45.7)	22 (40.0)	36 (46.8)	-
tumour localisation	*n* (%)	*n* (%)	0.597	*n* (%)	*n* (%)	*n* (%)	<0.001
right	64 (36.0)	48 (37.8)	-	28 (60.9)	11 (20.0)	25 (32.5)	-
transverse	19 (10.7)	13 (10.2)	-	6 (13.0)	5 (9.1)	8 (10.4)	-
left	44 (24.7)	36 (28.3)	-	4 (8.7)	17 (30.9)	23 (29.9)	-
rectosigmoideum	32 (18.0)	23 (18.1)	-	6 (13.0)	10 (18.2)	16 (20.8)	-
rectum	19 (10.7)	7 (5.5)	-	2 (4.3)	12 (21.8)	5 (6.5)	-
grade	*n* (%)	*n* (%)	0.998	*n* (%)	*n* (%)	*n* (%)	<0.001
NA, in situ	7 (3.9)	5 (3.9)	-	0 (0.0)	3 (5.5)	4 (5.2)	-
1	18 (10.1)	12 (9.4)	-	1 (2.2)	5 (9.1)	12 (15.6)	-
2	102 (57.3)	73 (57.5)	-	18 (39.1)	37 (67.3)	47 (61.0)	-
3	51 (28.7)	37 (29.1)	-	27 (58.7)	10 (18.2)	14 (18.2)	-
AJCC stage	*n* (%)	*n* (%)	0.968	*n* (%)	*n* (%)	*n* (%)	0.136
0	8 (4.5)	6 (4.7)	-	0 (0.0)	3 (5.5)	5 (6.5)	-
I	31 (17.4)	26 (20.5)	-	2 (4.3)	12 (21.8)	17 (22.1)	-
II	66 (37.1)	45 (35.4)	-	21 (45.7)	19 (34.5)	26 (33.8)	-
III	48 (27.0)	34 (26.8)	-	16 (34.8)	12 (21.8)	20 (26.0)	-
IV	25 (14.0)	16 (12.6)	-	7 (15.2)	9 (16.4)	9 (11.7)	-
tumour pathologic stage	*n* (%)	*n* (%)	0.979	*n* (%)	*n* (%)	*n* (%)	0.007
pTis	8 (4.5)	6 (4.7)	-	0 (0.0)	3 (5.5)	5 (6.5)	-
pT1	11 (6.2)	10 (7.9)	-	0 (0.0)	5 (9.1)	6 (7.8)	-
pT2	32 (18.0)	24 (18.9)	-	2 (4.3)	12 (21.8)	18 (23.4)	-
pT3	115 (64.6)	79 (62.2)	-	42 (91.3)	30 (54.5)	43 (55.8)	-
pT4	12 (6.7)	8 (6.3)	-	2 (4.3)	5 (9.1)	5 (6.5)	-
regional lymph nodes pathologic stage	*n* (%)	*n* (%)	0.618	*n* (%)	*n* (%)	*n* (%)	0.041
pN0	109 (61.2)	79 (62.2)	-	23 (50.0)	36 (65.5)	50 (64.9)	-
pN1	46 (25.8)	36 (28.3)	-	13 (28.3)	10 (18.2)	23 (29.9)	-
pN2	23 (12.9)	12 (9.4)	-	10 (21.7)	9 (16.4)	4 (5.2)	-
synchronous distant metastasis	*n* (%)	*n* (%)	0.846	*n* (%)	*n* (%)	*n* (%)	0.722
M0	153 (86.0)	111 (87.4)	-	39 (84.8)	46 (83.6)	68 (88.3)	-
M1	25 (14.0)	16 (12.6)	-	7 (15.2)	9 (16.4)	9 (11.7)	-
MSI/MSS	*n* (%)	*n* (%)	1	*n* (%)	*n* (%)	*n* (%)	<0.001
MSI	27 (15.2)	19 (15.0)	-	16 (34.8)	4 (7.3)	7 (9.1)	-
MSS	110 (61.8)	81 (63.8)	-	22 (47.8)	37 (67.3)	51 (66.2)	-
NA	41 (23.0)	27 (21.2)	-	8 (17.4)	14 (25.4)	19 (24.7)	-
*BRAF*	*n* (%)	*n* (%)	1	*n* (%)	*n* (%)	*n* (%)	0.022
*BRAF* wt	77 (43.3)	53 (41.7)	-	17 (37.0)	27 (49.1)	33 (42.9)	-
*BRAF* mut	12 (6.7)	9 (7.1)	-	7 (15.2)	1 (1.8)	4 (5.2)	-
NA	89 (50.0)	65 (51.2)	-	22 (47.8)	27 (49.1)	40 (51.9)	-
*KRAS*	*n* (%)	*n* (%)	1	*n* (%)	*n* (%)	*n* (%)	0.839
*KRAS* wt	24 (13.5)	17 (13.4)	-	7 (15.2)	8 (14.5)	9 (11.7)	-
*KRAS* mut	13 (7.3)	9 (7.1)	-	5 (10.9)	4 (7.3)	4 (5.2)	-
NA	141 (79.2)	101 (79.5)	-	34 (73.9)	43 (78.2)	64 (83.1)	-
*NRAS*	*n* (%)	*n* (%)	1	*n* (%)	*n* (%)	*n* (%)	0.553
*NRAS* wt	37 (20.8)	26 (20.5)	-	11 (23.9)	12 (21.8)	14 (18.2)	-
*NRAS* mut	2 (1.1)	1 (0.8)	-	1 (2.2)	1 (1.8)	0 (0.0)	-
NA	139 (78.1)	100 (78.7)	-	34 (73.9)	42 (76.4)	63 (81.8)	-

CRC—colorectal cancer, TMS—tumour microbial subtypes, SD—standard deviation, NA—not available, pT—tumour pathologic stage, pTis—tumour in situ, pN—regional lymph nodes pathologic stage, M—synchronous distant metastasis, MSI—microsatellite instability MSS—microsatellite stable, wt—wild type, mut—mutation.

**Table 2 cancers-13-04799-t002:** Summary of rank regression results (*p* < 0.05) associating microbiome of the three different sample types with the clinical variables. **Bold text** denotes genera significant at FDR < 0.1, text underlined by a solid line denotes that the association was validated in an independent dataset, marked by the superscript number (^1^ Feng et al. [30]; ^2^ Dejea et al. [31], * previously published association [40,66,68,69,70,71,72], see Discussion and Appendix A). Up and down arrows denote increase or decrease in abundance, respectively.

Regression Covariate	Effect/Contrast	Tumour Mucosa	Visually Normal Mucosa	Stool
grade	increasing grade	↑ ***Fusobacterium*** *,***Campylobacter****, ***Leptotrichia***, *Peptoclostridium*, *Mogibacterium* *	-	-
		-	↓ Unassigned genus from order *Opitutae vadin HA64*	-
location	right-sided/transverse vs left-sided and rectum/rectosigmoid	↑ *Holdemania*, *Selenomonas 4, Clostridium sensu stricto 1*, *Alloprevotella*	↑ ***Selenomonas 3*, *Selenomonas*, *Treponema 2***	-
		↓ ***Bifidobacterium*** *, ***Christensenellaceae R-7 *****group****^2^**, ***Ruminococcaceae UCG-013****, Fusicatenibacter*	↓ ***Lachnospira, Bifidobacterium, Coprococcus 1, Christensenellaceae R-7* group**	-
	right-sided/transverse vs left-sided	↑ *Campylobacter*, *Alloprevotella*	-	-
		↓ Family *XIII AD3011* group, *Coprococcus 1*	-	
	right-sided/transverse vs rectosigmoid/rectum	↑ *Oribacterium, Fretibacterium*	-	-
		-	↓ **[*Eubacterium*] *ventriosum* group**	
grade*location interaction	low-graded; right-sided/transverse	↑***Ruminococcaceae UCG-010*, uncultured bacterium from *Clostridiales vadinBB60* group**	-	↑ **Unassigned genus from order** ***Opitutae vadin HA64*, *Porphyromonas***
	grade 2; left-sided	↓***Coprococcus 2, Ruminiclostridium 6*, [*Eubacterium*] *ventriosum* group, *Incertae Sedis* from *Lachnospiraceae* family**	↓ ***Gemella*, *Corynebacterium 1***	↓ ***Ruminiclostridium 6***, *Coprococcus 2*
	grade 2; rectosigmoid/rectum	-	↑ ***Veillonella***	↑ ***Veillonella***
		↓ ***Methanobrevibacter*, *Dielma*, *Victivallis***	↓ ***Methanobrevibacter*, an** **uncultured genus from** **the *Peptococcaceae* family**	↓ ***Victivallis*, *Ruminiclosridium 6*, *Lachnospiraceae UCG-005*,** **an unassigned genus from order *Mollicutes* RF9**
	grade 3; right-sided/transverse	↑ ***Prevotella*, *Selenomonas*, *Selenomonas 3***	-	-
	grade 3; left-sided	-	↑ ***Eisenbergiella*, *Leptotrichia*, *Escherichia*-*Shigella*, *Veillonella***	↑ ***Veillonella, Prevotella 7***
		↓ ***Coprococcus 2, Ruminiclostridium 6, [Eubacterium] ventriosum* group, *Incertae Sedis* from *Lachnospiraceae* family, *Odoribacter***	↓ ***Gemella*, *Corynebacterium 1***	↓ ***Coprococcus 2***
	grade 3; rectosigmoid/rectum	↑ ***Lachnospira***	↑ ***Veillonella***	↑ ***Prevotella, Prevotella 7***
		↓ ***Methanobrevibacter, Dielma, Victivallis***	↓ ***Methanobrevibacter, Eisenbergiella*, an** **uncultured genus from** **the *Peptococcaceae* family**	↓ ***Lachnospiraceae UCG-005*,** **unassigned genus from order *Mollicutes RF9***
AJCC stage	III–IV vs 0–II	↑ *Peptoclostridium*	-	↑ ***Akkermansia***
		-	↓ *Gelria*	-
Tumour pathologic stage	pT 3–4 vs pTis-2	↑ ***Peptoclostridium, Gemella, Campylobacter, Parvimonas***	↑ ***Peptoclostridium, Escherichia-Shigella*, an unassigned species from *Ruminococcaceae***	↑ *Escherichia-Shigella*
		↓ ***Coprobacter****, Intestinimonas, Ruminococcaceae UCG-009, Oscillospira, Cloacibacillus*	↓ ***Intestinimonas, Ruminococcaceae UCG-009, Holdemanella, Coprobacter****, Gelria,* an uncultured genus from the *Christensenellaceae* family	↓ ***Prevotella 6****, Ruminococcaceae UCG-**011* ^1^
Regional lymph nodes stage	N1–2 vs N0	↑ *Peptoclostridium*	-	↑ *Peptococcus, Campylobacter, **Akkermansia*** **, Selenomonas, Porphyromonas ***, Streptococcus, Oscillospira*
		↓ *Prevotellaceae UCG-001,* uncultured *Fusobacterium sp.* from family *boneC3G7*	↓ [*Eubacterium*] *hallii* group	↓ Faecalibacterium, Ruminiclostridium, Dorea ^1^, Lachnospiraceae FCS020 group
Synchronous distant metastasis	present vs absent	↑ *Porphyromonas, Streptococcus, Ruminococcaceae UCG-005*	↑ *Akkermansia*	↑ **uncultured genus from** ***Erysipelotrichaceae* family**, *Akkermansia, Coprococcus 1, Solobacterium*
		-	↓ *Gelria*, [*Eubacterium*] *brachy* group, uncultured genera from *Christensenellaceae* family, *Gordonibacter*, *Fretibacterium*	↓ *Selenomonas,* *Ruminococcaceae UCG-004*

FDR-false discovery rate.

## Data Availability

Sequencing data were uploaded to the European Nucleotide Archive under accession number PRJEB35990.

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
