# Peer review of "Colorectal Tumour Mucosa Microbiome Is Enriched in Oral Pathogens and Defines Three Subtypes That Correlate with Markers of Tumour Progression"

_cancers, 2021, doi:10.3390/cancers13194799_

Round 1

Reviewer 1 Report

The revised article and comments are not acceptable.
Rejected

Author Response

We have no additional answers to provide other than we have already provided.

Reviewer 2 Report

The authors have satisfied my concerns 

Author Response

We are happy that we addressed all the reviewer’s concerns. In the current version we additionally corrected some spelling mistakes and typos.

This manuscript is a resubmission of an earlier submission. The following is a list of the peer review reports and author responses from that submission.

Round 1

Reviewer 1 Report

Dear authors I was impressed by the enormous amount of work performed and by the interesting design of your article, but finally it was analyze in a very confusing way with a not common statistically significance value. I am afraid that your article could not be evaluated in the proposed version and need a complete statistical reanalysis. Publishing an article like this one with the statistical pitfuls present, it give to the reader the impression of statistical significance values of a high numbers of genera where it is really only a trend or nothing.

  1. Page 4, line 170: the number of analyzed reads of the samples vary too much, around one hundred times ".....ranged from 2968 to 239116, with median of 44371 and mean of 53074 reads per sample." Usually it is safer to analyze data with  differences  not exceding than 10 times. In addition, the alpha and beta diversity indexes analysis is usually performed after data rarefaction to the minimum library size (in this case 2968 reads), thus loosing an enormous amount of informations. It is possible that low counts sample have not been considered in the analysis, but if this approach has been done it has not been reported.
  2.  Page 5, line 210: The statistical significance was considered arbitrarily 0.1 "Results were considered significant at FDR<0.1." This is not acceptable, because if a reader do not read carefully the article or read the abstract only is convinced that the taxa have a correct statistical significance, but that it is not.
  3. Page 7, line249-251. "Overall, in all the 483 samples we identified 5449 ASVs, of these, 4800 ASVs in the 127 triplet samples. The QIIME assigned species only to 48 ASVs, hence we also performed manual BLAST search to the SILVA database. The number and frequency of species definition is very low (less than 1%), probably since the total reads analyzed were not appropriate in numbers. 
  4. Figure 2A, line 286. "Vertical dashed line represents median relative abundance of all 264 detected genera (med=0.24%)." Based on biology this threshold line does not make any sense.
  5. Supplementary Materials, line133-134: "It is to be emphasized that the mean abundance per genera in the case of 264 genera is 100/264=0.379%. this again based on biology this threshold line does not make any sense.
  6. Figure S1 Panel B. The graphs scale is not correct as well as probably all the panels were wrongly attached.

Minor issues.

Page 4 line 184-185: "The threshold of 9 reads represents 0.03% taxa abundance in the sample with the least number of reads (2968)." It is wrong, since it is 10 times more  (0.3%)

Reviewer 2 Report

The manuscript submitted by Budinská investigated links between the gut microbiome and colorectal cancer by 16S rRNA gene sequencing of paired tumor mucosa, adjacent visually normal mucosa and stool swabs of 178 patients with CRC. The author identified 264 genera in these samples and found many varied significantly in abundance across sample types. Significantly, they found the bacteria residing on tumor-mucosa were dominated by genera belonging to oral pathogens. The author also found that many microbiome abundance associated with clinical variables (tumor localization, TNM staging, BRAF mutation and MSI status, KRAS,NRAS). In addition, the Author stratified CRC patients into three subtypes by tumor microbiome. Similarly, these tumour microbial subtypes are also significantly associated with many clinical variables such as tumor grade, primary tumor sidedness and TNM staging, BRAF mutation and MSI status.

This work has a large sample size (178 CRC patients). More importantly, they collected patients’ intestinal mucosa where gut microbiota directly lives. Mucosa microbiome can directly reflect the characteristic of gut microbiota in a pathologic state. The author made a systematical analysis between microbiome abundance and clinical variables. Moreover, the author defined three tumour microbial subtypes based on tumor microbiome, which is interesting and original, and also analyzed their correlation with clinical variables.

Major ones:

This work collected a large number of precious samples and made systematical analysis, but it was lack of research depth and failed to stress the main points due to investigated too many factors. This work primarily limits to the correlation analysis between microbiome and clinical parameters, and are lack of in-depth exposition and discussion of disease mechanisms. For example, though without healthy controls, there are many CRC patients with stage 0-IV, thanks to the large sample size. Although this work analyzed the correlation between microbiome and the progression of CRC, it does not delve deep into how the dysbacteriosis involve in the progression of CRC or propose a possible therapeutic target via targeting gut microbiota, which may have higher research value and be more interest to audiences. Finally, as the authors said, the paper does not provide definite answers.

Other comments:

  1. Inclusion criteria need to be described in more detail. Which of the patients could be recruited in and which should be excluded from the study.
  2. Some paragraphs (eg. 3.2.1) lack p-value or FDR, which should be added next to the data with statistical significance.
  3. The testing power (FDR<0.1) is too low for multiple hypothesis testing.
  4. Line 324, Bold text denotes genera significant at FDR<0.1. However, no genera were in bold in table 3.
  5. Functional profiling of the microbiome which was significantly associated with CRC stage is suggested to be supplemented in the manuscript. 

Reviewer 3 Report

The manuscript by Zwinsová et al provides a valuable resource of colorectal cancer associated bacteria from 178 patients and 3 sample types. They focus their attention to the differences between samples in-patient, rather than between patients, thus profiling bacteria that are enriched or depleted in the tumor mucosa, healthy adjacent mucosa or stool. The authors associate bacteria with relevant clinical data such as tumor stage and metastases presence. To the best of my knowledge, the main novelties in this work are the robust and systematic enrichment of oral bacteria observed and the identification of CRC subtypes the correlate bacterial communities with clinical values. Overall this is an interesting, though descriptive, study that adds validation to previous findings and some novel associations. 

Major points:
1. The amount of data presented is overwhelming. Is it possible to deliver the message more concisely with fewer data in the text, main tables and main figures? Keep it mostly in the supplementary data.
2. In their beta-diversity analysis, the authors report that the location of the tumor is a significant factor in the difference between the samples. What about the patients? I would expect patient-specific communities to be the most significant is separating the samples.
3. The authors can validate their results, in some capacity, by analyzing published data. For example, (PMID: 30936548 and PMID: 25489084) can be used to look at oral bacterial enrichment in CRC patients and tumor sidedness. 
4. To continue the previous point, the authors can analyze data sets of CRC patients vs healthy individuals and assess the significant of their stool biomarker findings, e.g., Akkermansia.
5. The discussion is too lengthy, it should be shortened and made more concise.
6. The authors raise an interesting point about metachronous tumors - are there any in their samples? So they could test their hypothesis.
7. The authors should acknowledge and discuss previous reports suggesting migration of oral microbes to the tumor, such as PMID: 32850497.

Minor points:
1. I feel like the main point of the study is the CRC subtypes associated with bacterial communities. These findings are not discussed in a clear fashion, in my opinion. 
2. What makes the tumor niche so attractive to oral bacteria? 
3. The second half of the abstract is too vague. 
4. How much extracted DNA was used as template for the 1st PCR?